# Significance of Intraplaque Hemorrhage for the Development of High-Risk Vulnerable Plaque: Current Understanding from Basic to Clinical Points of View

**DOI:** 10.3390/ijms241713298

**Published:** 2023-08-27

**Authors:** Atsushi Sakamoto, Kenichiro Suwa, Rika Kawakami, Alexandra V. Finn, Yuichiro Maekawa, Renu Virmani, Aloke V. Finn

**Affiliations:** 1CVPath Institute, Inc., Gaithersburg, MD 20878, USA; asakamo@hama-med.ac.jp (A.S.); rkawakami@cvpath.org (R.K.); alexandrafinn26@icloud.com (A.V.F.); rvirmani@hama-med.ac.jp (R.V.); 2Division of Cardiology, Internal Medicine III, Hamamatsu University School of Medicine, Hamamatsu 431-3125, Japan; k-suwa@hama-med.ac.jp (K.S.); ymaekawa@hama-med.ac.jp (Y.M.)

**Keywords:** atherosclerosis, vulnerable plaque, Intraplaque hemorrhage

## Abstract

Acute coronary syndromes due to atherosclerotic coronary artery disease are a leading cause of morbidity and mortality worldwide. Intraplaque hemorrhage (IPH), caused by disruption of Intraplaque leaky microvessels, is one of the major contributors of plaque progression, causing a sudden increase in plaque volume and eventually plaque destabilization. IPH and its healing processes are highly complex biological events that involve interactions between multiple types of cells in the plaque, including erythrocyte, macrophages, vascular endothelial cells and vascular smooth muscle cells. Recent investigations have unveiled detailed molecular mechanisms by which IPH leads the development of high-risk “vulnerable” plaque. Current advances in clinical diagnostic imaging modalities, such as magnetic resonance image and intra-coronary optical coherence tomography, increasingly allow us to identify IPH in vivo. To date, retrospective and prospective clinical trials have revealed the significance of IPH as detected by various imaging modalities as a reliable prognostic indicator of high-risk plaque. In this review article, we discuss recent advances in our understanding for the significance of IPH on the development of high-risk plaque from basic to clinical points of view.

## 1. Introduction

Atherothrombosis is a leading cause of morbidity and mortality worldwide [1]. Three plaque morphologies that trigger acute coronary syndrome (ACS) have previously been described based on human pathological observation, i.e., plaque rupture, plaque erosion and calcified nodule (Figure 1A–C) [2,3]. Plaque rupture is reported as the major cause (of ACS), according to observational studies including human autopsy series and/or clinical coronary imaging investigations [2,4,5]. Based on its morphological similarity, thin-cap fibroatheroma (TCFA) has been recognized as a likely precursor lesion of plaque rupture, often referred to as a “vulnerable plaque” (Figure 1D). With the recent development of clinical diagnostic imaging technologies, identification of TCFA in vivo clinical setting is becoming a possibility. Several prospective clinical studies have been conducted to prove the effectiveness of TCFA detection by coronary imaging modalities. To date, however, the positive predictive value of coronary unstable plaque detected by clinical diagnostic imaging modalities in terms of future cardiovascular event prediction (including primary and secondary prevention) is not precise enough to predict on a patient-based level who is at risk for a future event. Further research is needed in both pathology and clinical coronary imaging in order to guide more accurate clinical diagnoses.

Intraplaque hemorrhage (IPH) (Figure 1E) is thought to be a consequence of erythrocytes leakage from dysfunctional microvessels and subsequent local inflammatory cell accumulation, such as macrophages [6]. This is a highly regulated biological process resulting in one of the critical contributors to plaque destabilization, in part by causing a sudden increase in plaque volume [6,7]. With the development of clinical diagnostic imaging technologies, in vivo detection of IPH is becoming possible. In the field of carotid atherosclerosis, clinical evidence supports that IPH (mainly detected by magnetic resonance image [MRI]) is an independent risk factor for the future cerebrovascular events [8,9]. Clinically used coronary imaging modalities are in some instances able to detect coronary IPH in vivo and reported its importance in clinical outcome prediction [10,11,12]. Therefore, a new prediction model of coronary high-risk vulnerable plaque, including IPH findings, may improve prognostic accuracy of cardiovascular event prediction.

In this review paper, we discuss the recent understanding of IPH, one of the key features of the unstable plaque phenotype, and its potential for future coronary event prediction from basic to clinical points of view.

## 2. Intraplaque Hemorrhage: One of the Critical Features of Unstable Plaque

Since the 1930s, IPH has been described as one of the features of complex advanced atherosclerotic lesions [6,13,14]. In human pathology, IPH is observed mostly within and/or neighboring the necrotic core (Figure 1E). The rupture of Intraplaque leaky neovessels, mainly related to centripetal angiogenesis from the adventitia towards the plaque (i.e., vasa vasorum), has been considered as a trigger of IPH onset [6]. Generally, progressive outward vessel enlargement (positive remodeling) is observed in atherosclerotic lesions, which allows for the preservation of the lumen area up to a point [15]. Coronary artery stenosis, caused by atherosclerosis, only occurs once a plaque develops beyond 40% of cross-sectional area narrowing (i.e., Glagov’s phenomenon) [15,16,17]. Repetition of subclinical plaque rupture followed by non-occlusive luminal thrombus organization and IPH without luminal thrombosis are considered to be two major mechanisms of rapid luminal narrowing. Moreover, IPH is one of the key stimuli for destabilizing atheroma, which leads to plaque rupture accompanied by rapid progression of plaque burden. In preclinical studies, genetically modified atherogenic mice (e.g., ApoE^−/−^ and LDLR^−/−^) are usually applied as reliable in vivo models for human atherosclerosis. However, IPH, neovascularization and plaque rupture are not commonly detected in their atheromas. Thus, human pathology is still the major focus for research on IPH and neovascularization. In human pathologic sections, IPH can be generally detected by hematoxylin-eosin (H&E) or Masson trichrome stains, which are non-specific for hemorrhage detection. Additionally, further specific information can be obtained with special staining, such as Prussian blue (for iron deposition detection), and immunostaining with glycophorin A (a major sialoglycoproteins of red blood cell membrane) [6]. In a previously reported autopsy case series of sudden coronary death, evidence of IPH was identified by immunostaining for glycophorin A [18]. IPH incidence increased with the progression of arteriosclerotic lesions in the order of pathologic intimal thickening, early fibroatheroma, late fibroatheroma and TCFA. Of note, the frequency of the IPH was significantly greater in unstable atheroma phenotypes, such as late fibroatheroma and TCFA, than in early fibroatheroma [18].

## 3. Neovascularization as an Origin of Intraplaque Hemorrhage

Intraplaque vasa vasorum are reported to have incompetent barrier function of the endothelial layer. The disruption of endothelial integrity, which is generally maintained by intercellular junctions, results in the leakage of blood constituents from neovessels [19,20]. A prior electron microscopic study in human coronary atherosclerosis revealed the morphologic abnormality and impaired endothelial cell-cell junctions [21]. Since IPH is often observed adjacent to neovessels, it is thought that IPH may arise from these areas. A prior study revealed a defect in subendothelial mural cell support in adventitial vasa vasorum in human atherosclerotic coronary arteries, which is indispensable in the stabilization of vascular endothelial cells (ECs) and vascular structure to avoid leakage [21]. This indicates that sprouting and expanding immature plaque vasa vasorum are highly fragile, permeable and vulnerable to hemorrhage. Matrix metalloproteases and inflammatory cytokines, released from activated macrophages and mast cells, may lead to damage of neovessels and exacerbate IPH [22]. Vascular endothelial growth factor (VEGF) is key in the development of vascular permeability, which leads to the occurrence of IPH [23]. VEGF-A provokes ECs permeability via phosphorylation of VE-cadherin and internalization from cell membrane to cytosol, resulting in a loss of endothelial adherence junctions. Therefore, VEGF-A upregulation in plaques leads to increased permeability and promotes immature neovessels [24,25]. Maintaining ECs integrity may reduce the risk of IPH and subsequent plaque progression [26]. Microhemorrhages derived from immature neovessels may itself perpetuate the progress of angiogenesis through inflammatory mechanisms discussed below. Different active forms of matrix metalloproteinases (MMPs) are reported to be expressed in the proximity of neovascularized regions in vulnerable plaques, suggesting the possible involvement for microvascular leakage and impending IPH [27,28]. Neovessels are often composed by a single layer of ECs over the basement membrane without smooth muscle cell (SMC) support. Pathologic observation shows the frequent rupture of immature microvessels at close proximity to the site of the necrotic core. This can be due to a loss of matrix support in the necrotic core and the location between the fibrous cap and the base of plaque with more collagen. These areas may frequently be subjected to injuries caused by increased hemodynamic pressure changes in the vasculature.

Hemodynamic effects also contribute to the development of IPH. Prior clinical evidence supports the fact that arterial thrombotic occlusion tends to occur at or near the proximal region of the arterial stenosis [29]. It is known that vascular wall shear stress (WSS) is important for the atherosclerosis development [30]. The degree and/or spatial distribution of local WSS may change the development of plaque formation [31,32]. In the advanced stage of plaque (with large plaque burden), high shear stress (HSS) can be formed at the proximal end of the stenosis while low shear stress (LSS) may be formed at the distal portion [33]. It was previously reported that HSS plays a critical role for VEGF production [34] as well as endothelial nitric oxide (NO) synthesis [35,36,37]. Similar to VEGF, local NO mediates neoangiogenesis (i.e., shear-induced angiogenesis) in ECs [38] and enhances vessel permeability [39]. The HSS region of the plaque can be the susceptible area of neoangiogenesis [40]. Mechanical pressure changes may also provoke higher expression of MMPs from the macrophages adjacent to the neovessels. These events may synergistically increase the probability of IPH and subsequent vascular damage [41].

## 4. Intraplaque Hemorrhage, Cholesterol Crystal Accumulation and Iron-Derived Oxidative Stress

Erythrocyte membranes contain a large amount of free cholesterol, and it is thought that the accumulation of erythrocyte membranes in atheromas due to IPH is a critical source of the lipid component of the necrotic core and the formation of cholesterol crystals composed of unesterified cholesterol. Free cholesterol retention in cells and tissues lead monohydrate crystal formation, which is originally derived from intra-cellular hydrolyzation of endocytosed cholesterol esters [42] or comes directly from free cholesterol within cell membranes [43]. This Intraplaque cholesterol crystal accumulation may contribute extensively to necrotic core enlargement [18] and enhancement of Intraplaque inflammation by stimulate intra-cellular NLR family pyrin domain containing 3 (NLRP3) inflammasome pathway leading IL-1β production through interacting with several types of cells (e.g., macrophage, lymphocytes and neutrophils) [44]. Cholesterol crystals may also cause fibrous cap disruption via mechanical piercing [45], and subsequent luminal thrombus formation through the activation of local complement system after fibrous cap disruption [46]. Moreover, during erythrocyte lysis, which occurs in areas of IPH, free iron from free hemoglobin may contribute to iron-induced oxidative tissue damage. Fundamentally, the transformation of electrons between the ferrous (Fe^2+^) and ferric states (Fe^3+^) contributes to the formation of reactive oxygen species (ROS) through the Fenton reaction [47]. Local ROS production induces subsequent oxidative modification of biomolecules, such as lipoprotein, one of the critical inducers of atherogenesis. Paradoxically, areas of IPH show reduced ROS likely because highly specialized macrophages detoxify iron through intake of hemoglobin as will be discussed below although this mechanism can be overwhelmed [48,49]. Oxidized low density lipoprotein (ox-LDL) promote endothelial dysfunction [50] and assist in the cell formation of macrophages engulfed by membrane LDL receptors, which eventually leads to necrotic core expansion [51]. Therefore, IPH is one of the critical findings as a characteristic of unstable atheroma that can ultimately lead to plaque rupture and acute coronary thrombotic occlusion.

## 5. Diversity of Macrophage Phenotype at the Site of Atherosclerosis

It is known that macrophages differentiated from peripheral blood monocytes are crucial contributors to the progression of advanced plaques [52]. Monocytes attach to the damaged vascular endothelium and migrate into the subendothelial space after being attracted by inflammatory stimuli, such as modified lipoproteins (e.g., ox-LDL). Ox-LDL has greater atherogenicity as it stimulates macrophage cholesterol accumulation and subsequent foam cell formation. Many foam cells become apoptotic even though apoptotic cells are cleared efficiently by phagocytes in the early stage of lesion formation (i.e., efferocytosis). Similarly, defective efferocytosis gradually happens in advanced atheromas, leading to an accumulation of dead cells and their debris at the area of necrotic core [53,54]. Debris originated from dead foam cells and vascular SMCs lead to necrotic core formation consisting of highly pro-thrombotic components (i.e., tissue factor). Exposure of flowing blood to these components, by fibrous cap disruption, leads to thrombotic luminal occlusion that is central to the pathogenesis of acute myocardial infarction [55].

In our current understanding, the differentiation of monocytes to macrophages is determined by the local environment and the availability of cytokines. Gordon et al. described the theory that microenvironments direct monocytes into functionally distinct cell types [56]. Accordingly, activated macrophages are classified into the inflammatory M1 type (induced by type 1 T-helper cell (Th1) cytokines) and the anti-inflammatory, tissue repair M2 type (stimulated by Th2 cytokines) [57]. M1-type macrophages are differentiated by stimulation with inflammatory cytokines, such as IFN-γ, and lipopolysaccharides produce inflammatory cytokines and reactive oxygen species. Conversely, M2-type macrophages are differentiated by cytokines, such as IL-4 and IL-13; release anti-inflammatory molecules, including IL-10 as well as TGF-β; and contribute to tissue repair [57]. As mentioned, the role of cell death and lipids (e.g., ox-LDL) dominated the field of atherosclerosis study for many years; recent investigations revealed more detailed contributions of local inflammatory processes, including macrophage polarization, on atheroma development [58]. When the stimuli for monocytes are pro-inflammatory cytokines or ox-LDL, differentiation into M1-type macrophages can be provoked. In addition, monocytes can be differentiated into atypical subtypes when exposed to other specific stimuli. In the last decade, it has become clear that there are several subtypes in the group that was previously and collectively considered to be M2 type, and the aforementioned classification is likely too simple [59]. Since macrophages are located in different regions in the plaque (e.g., besides necrotic core, plaque shoulder, next to blood vessels, calcification, or site of IPH), different environmental stimuli, including cytokines and chemokines, may modulate their activation and polarization. Chinetti-Gbaguidi et al. outlined sub-populations of Intraplaque macrophages depending on their different activation stimuli. Accordingly, M2a macrophages are induced by IL-4 or IL-13, M2b by IL-1β or LPS, M4 by CXCL4, Mox by ox-LDL, and M(Hb) or Mhem by hemoglobin in the setting of IPH (Figure 2) [59].

Recent developments in single cell RNA-sequencing (sc-RNA seq) technology are unveiling further characteristics of macrophage heterogeneity in human atheromas. In the sc-RNA seq analysis of human atheroma in symptomatic and asymptomatic carotid artery disease, by Fernandez D. M. et al., six different subpopulations of macrophage in the carotid plaque were classified into six different subpopulations based on transcriptomic signature [61]. These data points further suggest the complexity of Intraplaque macrophage populations beyond cell surface marker protein expression.

## 6. Intraplaque Hemorrhage and CD163^+^ Alternative Macrophages

It is known that macrophages differentiated from peripheral blood monocytes are crucial contributors to the progression of advanced plaques [52]. Local stimuli drive phenotypic changes within macrophages, which distinguish them from classical foamy macrophages [48,62]. At the site of IPH, local oxidative stress leads to erythrocyte lysis [63]. Free hemoglobin, released from lysed erythrocytes, is immediately bound by plasma haptoglobin, which results in hemoglobin/haptoglobin (Hb:Hp) complexes [63]. Macrophages engulf Hb:Hp complexes via the CD163 membrane scavenger receptor [64,65]. In the forementioned macrophage diversity in the plaque, CD163 is reported as a surface marker of specific alternative macrophage subset, M(Hb) or Mhem [48,59,62]. Bengtsson et al. found a positive correlation between the CD163^+^ macrophages and plaque vulnerability in their pathologic study that used 200 human carotid plaques removed from 197 patients by endarterectomy [66]. Plaques obtained from patients with symptomatic carotid artery stenosis as well as plaques with higher vulnerability index showed greater expression of CD163 on both protein and mRNA levels [66]. Plaque vulnerability index was determined by the sum of lipids, macrophages and hemorrhage by glycophorin, a staining, divided by the sum of collagen and SMCs. Recent sc-RNA seq data of human atheroma in symptomatic carotid artery disease also demonstrated the macrophage cluster characterized by upregulation of genes responsible for iron metabolism and iron storage, representing alternatively activated macrophage with proatherogenic potential involved in clearance of iron derived from hemoglobin deposited in IPH site [61]. Along with these observations, in our pathologic study using post-carotid endarterectomy specimens, we found that the number of CD163^+^ macrophages in the atheroma was positively correlated with the atherosclerosis progression and the number of microvessels in the plaque (Figure 3) [67].

### 6.1. CD163^+^ Macrophages, Microvascular Permeability and Inflammation

In vitro experiments using human, peripheral blood mononuclear cells showed that VEGF production was enhanced from Hb:Hp-stimulated CD163^+^ macrophages (M (Hb)). Supernatants from M (Hb) increased EC capillary tube formation, vascular permeability, and the inflammatory response of the vascular endothelium [67]. These effects were shown to be directly related to the production of VEGF by M (Hb). Of note, the formerly mentioned iron-related cellular toxicity and ROS generation through the Fenton reaction was not seen in M (Hb) due to greater expression of the iron transporter protein ferroportin (FPN) and subsequent releasing of intra-cellular iron to the outside of cell in its ferric form which then may be bound by transferrin (Figure 4). Further, an in vivo experiment comparing atherosclerotic lesions in brachiocephalic arteries between genetically modified atheroprone model mice (i.e., ApoE^−/−^ and ApoE^−/−^CD163^−/−^) revealed a significantly lower atheroma burden, necrotic core size, lesion stenosis and micro angiogenesis in ApoE^−/−^CD163^−/−^ as compared to ApoE^−/−^ mice, implicating CD163 in atheroprogression. It was confirmed that the peritoneal macrophages from CD163^−/−^ mice had significantly lower VEGF production when cultured under hemoglobin stimulation than those of wild-type mice [67]. CD163^+^ macrophages at the site of IPH contribute to the plaque development and destabilization by releasing VEGF. Local VEGF upregulation enhances vascular permeability and inflammatory cell infiltration into the plaque, thereby promoting further bleeding from immature neovasculature [67].

### 6.2. CD163^+^ Macrophages and Vascular Calcification

Coronary artery calcification is known to be a manifestation of advanced atherosclerotic disease. The detailed characteristics of calcification are related to the specific phenotypes of atheroma [69]; i.e., dense calcified plaques are related to stable coronary disease [70] while micro and fragmented calcification are associated with unstable plaques, which may lead to ACS [71].

Recently, we further elucidated the contribution of IPH to the development of less-calcified vulnerable plaque phenotypes through interaction between CD163^+^ macrophages and Intraplaque vascular SMC [68]. Pathologic assessment of human arteries revealed the inverse correlation between the distribution of CD163^+^ macrophages and calcification. In vitro experiments using vascular SMCs cultured with a supernatant of M (Hb) showed anti-calcific effect while arteries from ApoE^−/−^CD163^−/−^ mice showed greater vascular calcification compared with that from ApoE^−/−^ mice. M (Hb) supernatant-exposed SMCs showed activation of the NF-κB pathway. Blocking this signaling pathway attenuated the anti-calcific effect of M (Hb) on SMCs. M (Hb) altered calcification through NF-κB-induced transcription of hyaluronan synthase (HAS), an enzyme that catalyzes the formation of the extracellular matrix glycosaminoglycan and hyaluronan within SMCs. M (Hb) supernatants enhanced HAS production in SMCs, while knocking down HAS attenuated its anti-calcific effect. NF-κB blockades in ApoE^−/−^ mice reduced hyaluronan and increased calcification in the plaque. In human arteries, hyaluronan and HAS were increased in areas of CD163^+^ macrophage presence. These findings show an important mechanism by which CD163^+^ macrophages (in the plaque) inhibit extensive calcification through NF-κB-induced HAS augmentation in SMC, promoting the high-risk vulnerable plaque [68].

Taken together, at the sites of IPH: (1) the accumulation of free cholesterol in the necrotic core due to erythrocyte membrane deposition (Figure 4A), (2) lipoprotein oxidation by ROS caused by biological reaction of free iron released from free hemoglobin (Figure 4B) and (3) the inflammatory response of macrophages to hemoglobin released from erythrocytes interacting with several types of cell; e.g., ECs and SMCs (Figure 4C,D), synergistically contribute to increase necrotic core size, fragility of the plaque and may result in further cardiovascular complications.

## 7. Intraplaque Hemorrhage and Clinical Prognosis Examined by Diagnostic Imaging Modalities

The pathological findings of IPH site within the plaque undergo constant evolution over time (e.g., acute hemorrhagic phase, thrombus organization, local inflammatory cells accumulation, micro angiogenesis, and replacement with fibrous tissue in chronic phase). In addition, since most IPH often occurs within or beside the necrotic core, necrotic tissue and lipid components might coexist with various degrees of mixture at IPH sites, affecting the prolonged local healing time sequence. It might be difficult to recognize IPH as a uniformly identifiable lesion by clinical diagnostic imaging modalities. In this regard, to date, magnetic resonance imaging (MRI) in carotid atherosclerosis is the most established diagnostic tools for the detection of IPH, which, based on iron detection with T1-weighted sequences, gives an Intraplaque hyperintense signal, whereas other components are detected as isointense or hypointense signals [72]. Particularly, T1-weighted gradient echo sequences (T1w-GRE) are frequently used to visualize the IPH as a lesion with high signal intensity [73,74]. The other important plaque component of the lipid-rich necrotic core, often adjacent to IPH, is visualized as the hypointense lesion in the plaque on proton density-weighted (PDW) fast spin echo or PDW-echo planar imaging and T2-weighted echo planar imaging sequences [73,74]. Calcification in the plaque is detected as hypointense lesions in any of the aforementioned sequences.

Several prospective clinical trials have shown that the presence of MRI-defined IPH in carotid atheroma is a robust predictor for future risk of ipsilateral ischemic stroke and/or transient ischemic attack (TIA) (Table 1). Takaya et al. reported a prospective observational study including 29 patients with carotid atherosclerosis with or without IPH findings (14 vs. 15 subjects) by baseline MRI assessment [7]. Over the 18 months follow-up, the percent change in wall volume (6.8% vs. −0.15%, *p* = 0.009) and lipid-rich necrotic core volume (28.4% vs. −5.2%, *p* = 0.001) was significantly greater in IPH (+) vs. in IPH (−) group [7]. Another carotid MRI study also reported greater increase in lipid content (1.2 ± 2.5%/year vs. −1.0 ± 2.2, *p* = 0.006) and greater decreases in lumen area (−0.4 ± 0.9 mm^2^/year vs. 0.3 ± 1.4, *p* = 0.033) in plaques with IPH during two-years follow up in patients on continued lipid-lower therapy [75]. These findings lie in line with the observation in basic research that showed cholesterol accumulation in the necrotic core of the plaque with IPH. According to the meta-analysis of eight clinical studies involving patients with symptomatic and/or asymptomatic carotid stenoses, the presence of IPH increases the risk of cerebrovascular events 5.69 times, with an annualized event rate of 17.7% with IPH and 2.4% without IPH [9]. Another meta-analysis by Gupta et al. also revealed MRI-derived IPH as a strong predictor of subsequent stroke/TIA (HR 4.59 [95% CI, 2.91–7.24]) [76]. In the recently reported Rotterdam study, 1349 patients (mean 72-yo) with asymptomatic carotid atherosclerosis diagnosed by carotid ultrasonography and no history of stroke or CAD were involved [74]. Evaluation of carotid plaque by MRI was conducted, and clinical events were prospectively assessed for mean follow-up periods of 5 years. During the follow-up, 51 stroke and 83 events of ischemic heart disease were observed. Regardless of the maximum thickness of atheroma and the presence of cardiovascular risk factors, the presence of IPH by MRI has been shown to be an independent predictor for the development of stroke and ischemic heart disease [74].

Other modalities, such as contrast-enhance computational tomography (CT), have also been reported to be able to distinguish IPH from lipid-rich and fibrous parts in carotid atheromas [77,78]. Saba et al. conducted a validation study comparing CT angiography and histopathologic findings for specimens obtained by carotid endarterectomies in 91 patients. Statistical association between IPH and low Hounsfield units (HU) was observed, and a threshold of 25 HU demonstrated the presence of IPH with a high sensitivity and specificity [77]. Another study, which tested the performance of carotid CTs, failed to demonstrate the utility of CT values to distinguish plaques with IPH from those without IPH due to significant overlap in the distribution of CT values [78].

As an alternative, a novel technique was developed to detect the co-existing neovascularization in high-risk plaque [79]. Serial CT acquisition at a short imaging interval, following contrast injection, was performed. Significantly lower washout was observed in plaques with high-risk CT features versus those without. At present, the ability of CT to discriminate IPH from other plaque components is not good enough when compared with that of the MRI [78]. For the better discrimination of IPH from other plaque components, more reliable digital algorithms for CT imaging analysis need to be developed.

Because coronary arteries and their atheromas are quite smaller than carotid arteries, it is also more difficult to analyze these vessels due to their movement with the heartbeat. Thus, determination of IPH in coronary arteries using imaging tools is tough to accomplish. Nevertheless, recent advancements of coronary imaging tools have been allowing us to detect likely IPH lesions in coronary atheromas. Noguchi et al. prospectively investigated whether coronary hyperintensity atheromas (HIP), as observed by non-contrast T1-weighted turbo field echo sequence images on cardiac MRIs, were predictors of future coronary ischemic events in 568 patients [10]. A pathologic validation study revealed HIP in cardiac MRI was presumed to be a mixture of lipid rich necrotic cores and IPH [11] (Figure 5). Coronary events were observed in 55 patients during the following up period of 55 months (median). In the HIP lesions, the incidence of coronary events was particularly high in the lesions with a greater plaque-to-myocardium signal intensity ratio (PMR), and a significant association with future events was observed [10]. The recent MRI sequence of T1-weighted images to detect HIP is a three-dimensional, whole-heart imaging technique of coronary atherosclerosis T1-weighted characterization (CATCH) [80,81]. Simultaneous acquisition of dark-blood plaque images by T1-weighted gradient echo sequence and bright-blood MR angiography for anatomical reference (fusion image) offers more accurate anatomical location of HIP.

It has recently been reported that intra-coronary plaque characterization by optical coherence tomography (OCT) is also useful to identify IPH lesions [12]. The co-location of low-intensity area without attenuation (LIA) and cholesterol crystals (CC), detected by OCT, may indicate IPH in the coronary plaque. In this study, Usui et al. reported that the prevalence of LIA + CC was 15.5% in 735 non-culprit lesions of PCI treated coronary arteries in 566 patients assessed by OCT [12]. In patients with LIA + CC, the incidence of major adverse cardiovascular events (MACE: cardiac death, myocardial infarction and revascularization due to irresponsible lesions) during the 3-years follow-up period was reported as 2.9% at the lesion and 15.6% at a patient level, respectively. The presence of LIA + CC detected by OCT is likely to be a part of the time course for the onset and healing of relatively large IPH lesions and was a strong predictor for future clinical events, OCT-detected TCFA, and minimum lumen area < 3.5 mm^2^. Taken together, IPH is one of the robust characteristics related to the prognosis of unstable atheroma.

**Table 1 ijms-24-13298-t001:** Identification of Intraplaque hemorrhage by imaging modalities and its clinical significance.

VSL	Authors/Study	Population	Country	Modality	Characteristics of IPH	No. of Pt (lesion)	Mean F/O (months)	Endpoint	Results	Interpretation
**CoA**	Usui E. et al. 2021 [12]	Pts who underwent PCI due to ACS or SAP. NCLs of PCI-treated VSL.	JPN	OCT	LIA + CC	566 (735)	30	MACE **	CC + LIA were found in 114/735 NCLs (15.5%). Untreated NCLs with LIA + CC had an increased risk for NCL-MACE (adjusted HR 3.09 [95% CI 1.27–7.50]).	An OCT-detected LIA + CC in an NCL was associated with subsequent NCL-MACE.
**CoA**	Noguchi T. et al. [10]	Pts with SAP underwent cardiac CT and MRI	JPN	MRI	HIP *	568	55	MACE **	HIP with PMRs ≥ 1.4 was the independent predictor of future coronary events (HR: 3.96 [95% CI 1.92–8.17]).	HIPs may represent a novel predictive factor.
**CA**	Rotterdam study [74]	Asy CA stenosis Pts detected by US w/o history of CAD and/or stroke	NLD	MRI	High intensity PL on 3D-T1W-GRE	1349	61.2	Stroke, cardiac death, CoA revascularization	MRI-derived IPH was associated with incident stroke and CHD (adjusted HR: 2.42 [95% CI: 1.30–4.50], and 1.95 [95% CI 1.20–3.14]).	IPH in the CA P is an independent RF for stroke and CHD. IPH can be a marker of plaque vulnerability.
**CA**	Saam T. et al. [9]	Meta analysis of 8 studies: Sym and/or Asy CA stenosis (30–99% stenosis)	USA, CAN, JPN, UK	MRI	(1) High intensity PL on 3D-T1W-GRE or (2) classification of IPH by levels of T1W, T2W/PDW and TOF intensities	689 (712)	19.6	Stroke, TIA	MRI-derived IPH was associated with an ∼6-fold higher risk for events (HR: 5.69 [95% CI 2.98–10.87]).	Presence of IPH on MRI strongly predicts cerebrovascular events.
**CA**	Gupta A. et al. [76]	Meta analysis of 9 studies: Sym and/or Asy CA stenosis (<50–79% stenosis)	USA, CAN, JPN, NLD, CHE, UK	MRI	(1) High intensity PL on 3D-T1W-GRE or (2) classification of IPH by levels of T1W, T2W/PDW and TOF intensities	678 (702)	20.2	Stroke, TIA	The HR for MRI-defined IPH as predictor of subsequent stroke/TIA were 4.59 [95% CI 2.91–7.24].	MRI-defined IPH is associated with risk for stroke or TIA. Plaque composition by MRI gives stroke risk information beyond degree of stenosis.
**CA**	Singh N. et al. [82]	Asy CA atherosclerosis Pts detected by US (50–70% stenosis)	CAN	MRI	High intensity PL on 3D-T1W-GRE	75 (98)	24.9	Stroke, TIA	MR-derived IPH was associated with an increased risk of cerebrovascular events (HR 3.59 [95% CI 2.48–4.71]).	MRI-defined IPH is associated with stroke/TIA in Asy CA stenosis. The absence of IPH at MRI may be a marker of plaque stability and of a lower risk of future event.
**CA**	Takaya N. et al. [83]	Asy CA atherosclerosis Pts detected by US (50–79% stenosis)	USA	MRI	Classification of IPH (fresh, recent, old) by levels of T1W, T2W/PDW and TOF intensities	154	38.2	Stroke, TIA	Presence of MRI-defined IPH (HR 5.2 [95% CI 1.6–17.3]) and larger mean IPH area (HR for 10 mm^2^ increase 2.6 [95% CI 1.4–4.6]) were associated with subsequent symptoms.	IPH by MRI is associated with future cerebrovascular events in patients with asymptomatic moderate carotid stenosis.

ACS = acute coronary syndrome, Asy = asymptomatic, CA = carotid artery, CABG = coronary artery bypass grafting, CAD = coronary artery disease, CAN = Canada, CC = cholesterol crystals, CI = confidence interval, CoA = coronary artery, CT = computed tomography, F/O = follow-up, HIP = high intensity plaque in non-contrast T1W MRI, HR = hazard ratio, IPH = Intraplaque hemorrhage, LIA = low-intensity area without attenuation, MACE = major adverse cardiac event, MI = myocardial infarction, MRI = magnetic resonance image, NCL = non-culprit lesion, OCT = optical coherence tomography, PCI = percutaneous coronary intervention, PDW = proton density-weighted, PL = plaque, PMR = plaque-to-myocardium signal intensity ratio, Pts = patients, RF = risk factor, SAP = stable angina pectoris, Sym = symptomatic, T1W-GRE = T1-weighted gradient echo, T2W = T2-weighted, TIA = transient ischemic attack, TOF = time of flight, US = ultrasonography, VSL = vessels, w/o = without. * HIP speculated as a mixture of lipid and IPH. ** MACE included cardiac death, MI and coronary revascularization with PCI or CABG.

At present, one of the possible technologies for detecting in vivo biological IPH beyond plaque morphology is the near-infrared autofluorescence (NIRAF) imaging [84]. Htun et al. reported autofluorescence in the NIR range is able to characterize IPH containing heme degradation products using a mouse model of IPH and human carotid endarterectomy samples [84]. Recently, a dual-modality OCT-NIRAF optical catheter system has been developed for the investigation of human coronary artery [85,86]. A pathology validation study by Kunio et al. clearly demonstrated the spatial co-localization of NIRAF signal and IPH features (positive staining for glycophorin A and ceroid) using 23 fresh human cadaver hearts, suggesting that NIRAF may provide additional, complementary information to morphologic imaging that could be used to identify high-risk coronary plaques, including IPH [87]. Further studies need to be done for elucidating the significance of in vivo IPH detection by clinically available imaging modalities and implications for future cardiovascular events prediction. Through evaluation in combination with other features of unstable plaque, such as thin fibrous cap, IPH may provide further improvement of predictive accuracy for clinical prognosis of vulnerable plaque.

## 8. Conclusions

Recent basic research has shown new insights on the manner in which IPH contributes to atheroma development and its vulnerability. Although lipid-lowering therapy has dramatically improved the prognosis of atherosclerotic disease, patients with LDL-C levels below the guideline-recommended target still have significant residual cardiovascular event rates [88]. In this regard, an understanding of non-lipid-oriented mechanisms of atherosclerosis development, such as IPH, is crucial in order to find another therapeutic target and/or novel way for risk stratification of atherosclerotic disease. Recent development of cardiovascular imaging modalities is allowing us to detect IPH in vivo and has been already shown supportive data for predicting future cardiovascular risk. Combining IPH with currently available imaging-based high-risk plaque features (e.g., positive remodeling, thin-fibrous cap, lipid-rich etc.), more accurate risk stratification strategies for “vulnerable plaque” detection can be achieved.

## Figures and Tables

**Figure 1 ijms-24-13298-f001:**
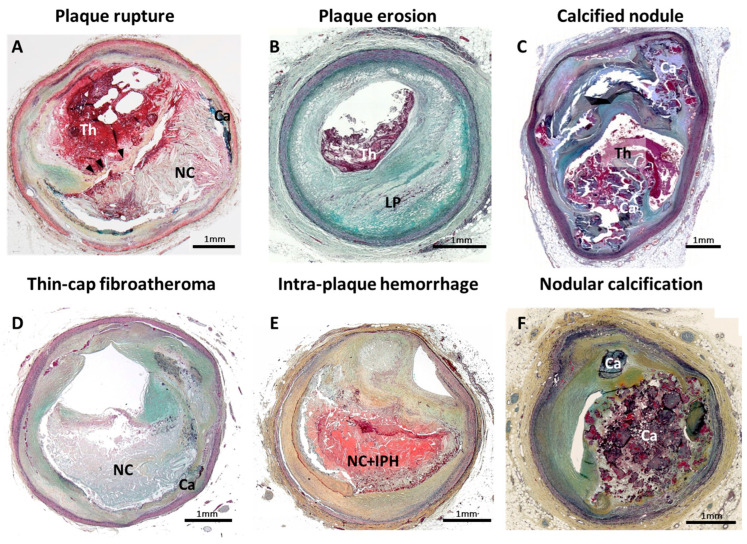
Pathology of human coronary artery morphologies associated with acute coronary syndrome. Pathologic classification of human coronary artery associated with acute coronary syndrome is shown. (**A**) Plaque rupture; (**B**) plaque erosion with underlying pathologic intimal thickening; (**C**) calcified nodule; (**D**) thin-cap fibroatheroma; (**E**) Intraplaque hemorrhage; and (**F**) nodular calcification. Ca = calcification, IPH = Intraplaque hemorrhage, LP = lipid pool, NC = necrotic core, Th = thrombus. Modified permission obtained from Sakamoto A. et al., US Cardiol Rev 2022 [3].

**Figure 2 ijms-24-13298-f002:**
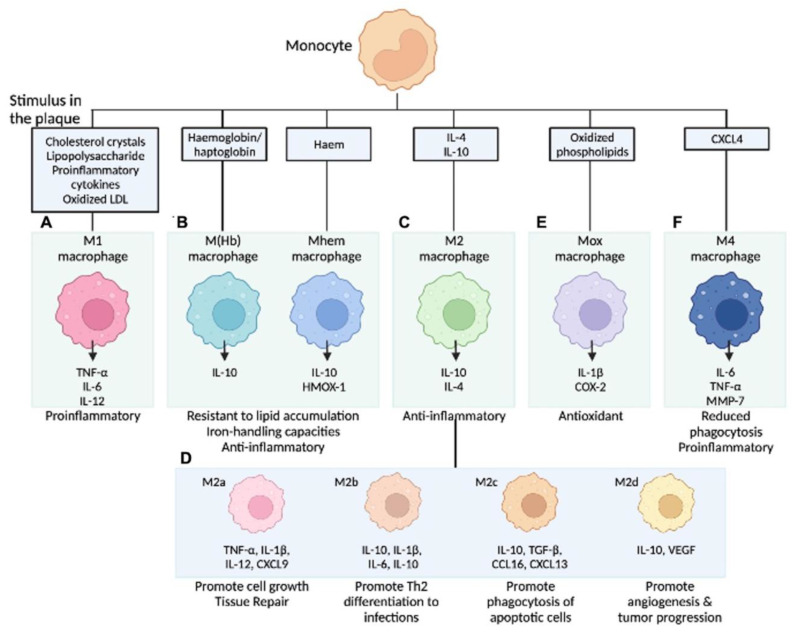
Macrophage subpopulations in atherosclerotic lesions. Stimuli present in atherosclerotic lesions drive the differentiation of monocytes towards different macrophage phenotypes. (**A**) M1 macrophages release pro-inflammatory cytokines; (**B**) M (Hb)/Mhem are associated with intraplaque hemorrhage; (**C**) M2 macrophages are anti-inflammatory and are less capable of lipid accumulation; (**D**) M2 is further categorized into four subtypes by activating molecules; (**E**) Mox macrophages are induced by oxidized phospholipids and increase the expression of antioxidant enzymes to protect oxidative stress; and (**F**) M4 macrophages express pro-inflammatory cytokines and have impaired phagocytic function. COX-2 = cyclooxygenase-2, CXCL4 = C-X-C motif chemokine 4, HMOX-1 = haem oxygenase (decycling) 1, LDL = low-density lipoprotein, LXR = liver X receptor, MMP-7 = matrix metalloproteinase-7, NFE2L2 = nuclear factor (erythroid-derived 2)-like 2, NF-κB = nuclear factor kappa-light-chain-enhancer of activated B cells, TLR = toll-like receptor, TNF = tumor necrosis factor. Modified permission obtained from Chinetti-Gbaguidi G. et al., Nat Rev Cardiol 2015 [59] and Kawai K. et al., Expert Rev Cardiovasc Ther 2022 [60].

**Figure 3 ijms-24-13298-f003:**
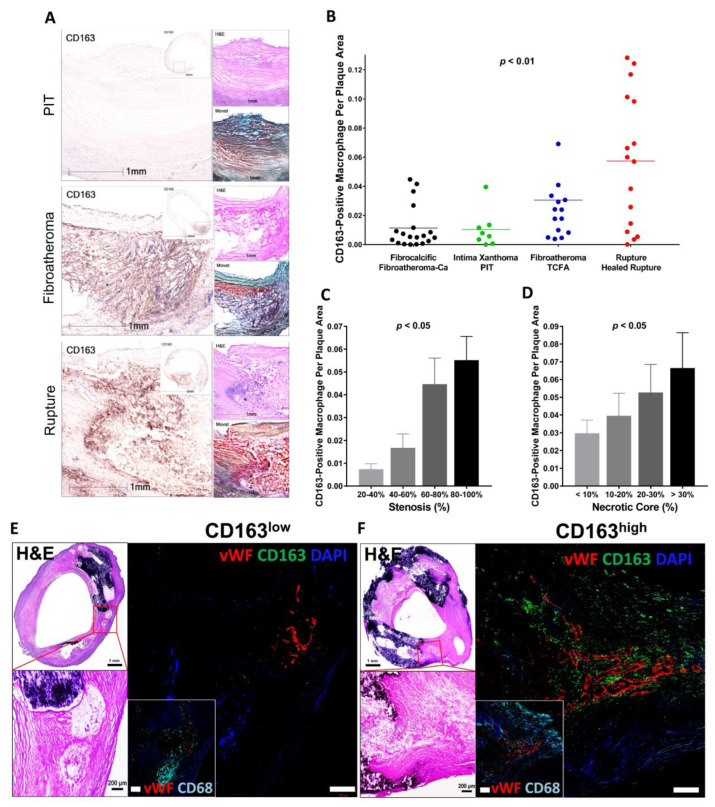
CD163^+^ macrophages are associated with progression of carotid atherosclerosis in humans. (**A**) Representative images of human carotid arteries with PIT, fibroatheroma and ruptured atherosclerotic plaques. High-magnification immunohistochemistry images of CD163 with low-magnification insets. H&E and Movat pentachrome stains are also shown. Scale bars: 1 mm and 5 mm (insets). (**B**–**D**) Correlation between CD163^+^ macrophages and human carotid plaque progression. (**B**) Human carotid plaques were classified as fibrocalcific or fibroatheroma with calcification (Fibroatheroma-Ca) (*n* = 19, black); intima xanthoma or PIT (*n* = 8, green); fibroatheroma or TCFA (*n* = 14, blue); and ruptured or healed rupture (*n* = 16, red), with the corresponding percentage of CD163^+^ macrophages per plaque area. (**C**) Correlation between CD163^+^ macrophages and the percentage of stenosis. The percentage of stenosis was categorized as follows: 20–40% (*n* = 5), 40–60% (*n* = 14), 60–80% (*n* = 23), and 80–100% (*n* = 28). (**D**) Correlation between CD163^+^ macrophages and the percentage of necrotic core area. The percentage of necrotic core area was classified as: <10% (*n* = 32), 10–20% (*n* = 19), 20–30% (*n* = 9), and >30% (*n* = 10). (**E**,**F**) Human plaques obtained by carotid endarterectomy were examined by histology and immunofluorescence. Images were acquired by confocal microscopy (**E**,**F**). In (**E**), as explained in the text, areas from fibroatheroma lesions containing foam cell (CD163^−^ [green], CD68^+^ [cyan]) macrophages (i.e., low CD163) and M (Hb) macrophages (CD163^+^, CD68^+^ [i.e., high CD163]) were immunostained using antibodies against vWF antigen for detection of microvessels. Nuclei were counterstained using DAPI (blue). Note that calcification is seen as dense areas of dark purple. Adjacent low- and high-magnification images of H&E-stained sections show the corresponding regions of angiogenesis. Scale bars: 1 mm and 200 μm. Results are presented as the mean or the mean ± SEM. For multiple group comparisons, 1-way ANOVA was used. *p* values shown in (**B**–**D**) were determined by 1-way ANOVA. Permission obtained from Guo L. et al., J. Clin. Invest. 2018 [67].

**Figure 4 ijms-24-13298-f004:**
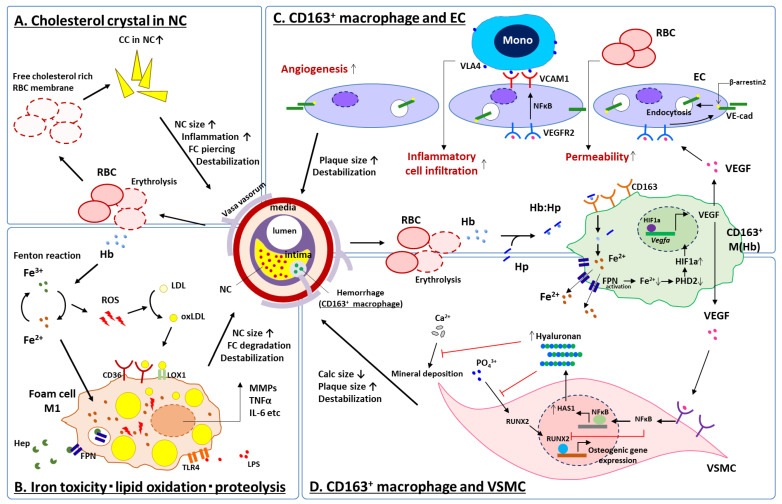
Schematic representation for current understanding of IPH on high-risk plaque development. At the site of IPH, accumulation of free cholesterol in the necrotic core due to erythrocyte membrane deposition (**A**), lipoprotein oxidation by ROS caused by biological reaction of free iron released from free hemoglobin (**B**) and inflammatory response of CD163^+^ macrophages to hemoglobin released from erythrocytes interacting with ECs (**C**) and VSMCs (**D**), synergistically contribute to increase necrotic core size, fragility of the plaque, and may result in further cardiovascular complications. CC = cholesterol crystal, CD = cluster differentiation, EC = vascular endothelial cell, FC = fibrous cap, FPN = ferroportin, HAS1 = hyaluronan synthase 1, Hb = hemoglobin, Hep = hepcidin, HIF1a = hypoxia inducible factor 1 alpha, Hp = haptoglobin, IL-6 = interleukin 6, LDL = low density lipoprotein, LOX1 = lectin-like oxidized LDL receptor 1, LPS = lipopolysaccharide, NFκB = nuclear factor kappa B, oxLDL = oxidized LDL, MMP = matrix metalloproteinase, Mono = monocyte, NC = necrotic core, PHD2 = prolyl hydroxylase domain-containing protein 2, RBC = red blood cell, ROS = reactive oxygen species, RUNX2 = runt-related transcription factor 2, TLR4 = toll-like receptor 4, TNFα = tumor necrosis factor alpha, VCAM1 = vascular cell adhesion molecule 1, VE-cad = vascular endothelial cadherin, VEGF = vascular endothelial growth factor, VEGFR2 = VEGF receptor 2, VLA4 = very late antigen 4, VSMC = vascular smooth muscle cell. Modified permission obtained from Guo L. et al., J. Clin. Invest. 2018 [67], and Sakamoto, A. et al., JCI insight 2023 [68].

**Figure 5 ijms-24-13298-f005:**
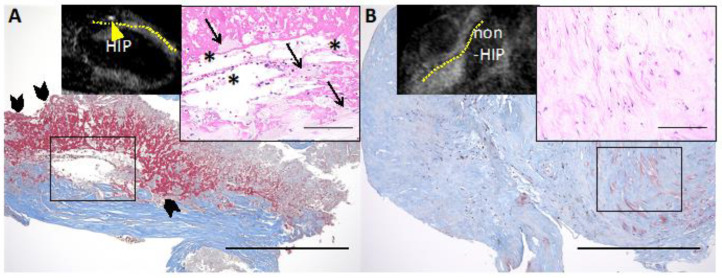
Representative histological images of HIP and non-HIP lesions in the coronary plaque detected by cardiac MRI. (**A**) Non-contrast T1-weighted magnetic resonance imaging (T1WI) shows high-intensity plaque (HIP) (plaque-to-myocardium signal intensity ratio (PMR) = 4.42, yellow arrowhead; the yellow dotted line indicates the coronary artery). The HIP lesion contains fibrous tissue with abundant atheroma components, such as cholesterin crystals (solid asterisks) and foamy macrophages (solid arrows) with red blood cells. (**A**) A large amount of fibrin deposition (solid arrowheads) is seen adjacent to the atheroma components. (**B**) Noncontrast T1WI shows non-HIP (PMR = 0.97). The non-HIP lesion demonstrates fibrous connective tissue. Low-power images of Masson’s trichrome stain (scale bars: 500 μm) and high-power images of hematoxylin and eosin stain (insets, scale bars: 100 μm). Permission obtained from Uzu K. et al., JACC Cardiovasc Imaging 2021 [11].

## Data Availability

Not applicable.

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
