# Peer review of "Significance of Intraplaque Hemorrhage for the Development of High-Risk Vulnerable Plaque: Current Understanding from Basic to Clinical Points of View"

_ijms, 2023, doi:10.3390/ijms241713298_

Round 1

Reviewer 1 Report

The review proposes to discuss the recent understanding of IPH and its potential for future coronary event prediction in a clinical setting from basic to clinical points of view.  This is a very important topic given the high residual risk of CVD events upon lipid lowering treatment and impact of IPH on plaque development and stability. Covering the background understanding of how IPH plays a role in plaque development and instability provides a nice mechanistic link to as to why IPH may predict future events.  

 There is a lot of good information in the paper, and the figures are very useful and a strength of the paper.  However, I feel that the overall flow of the paper, and that covered in each section needs addressing. The flow starts appropriately with section 2 addressing the clinical link between IPH and unstable plaque. Then section 3, steps back to explain how IPH forms. But then, instead of outlining the mechanistic ways in which IPH contributes to plaque development and instability, we have a section (section 4) on macrophage diversity in the plaque. It is debatable whether (or what from) this section is needed. Section 5 is key as it is outlining how IPH contributes to plaque development and instability. We see Figure 4 in this section -which is a nice way of putting all the information together. Currently however, we find that the information relating to the boxes in Figure 4, A- D is spread throughout the manuscript in different sections and using different headings than those used in Figure 4. The flow of the paper would be more coherent if the mechanisms by which IPH contributes to plaque instability were all together and aligned more with figure 4. Section 6, is important to bring us back to the clinical setting and asking can we detect IPH clinically.  

Section 2: This section describes the importance of intra- plaque hemorrhage as a feature of an unstable plaque. The setting up of the clinical importance of IPH is key, but the section does start to move into mechanism by which IPH contributes to plaque instability though formation of cholesterol crystals. This could move to section 5, which covers the mechanistic role of IPH for, indeed, step A in figure 4 is about CC.

Section 3:  This section covers IPH formation. This is appropriate. However, the section could be polished to read smoother, as it repeats the concept that hindering endothelial integrity leads to leaky vessels that increases IPH risk. The words leaky or leakage are repeated often: line 125, 130, 133, 134 (permeability), 143, 145 (permeability) 150 and 153.

 Section 4:  This section addresses macrophage diversity in the plaque outlining the understanding that has been gleaned over the ‘last several years’ (line 193). However, the M(Hb) or Mhem macrophage, was described in the plaque over 10 years ago. The diversity of macrophage phenotypes in the plaque should be well known by the reader. I do appreciate that the journal has a broader readership than the CVD or macrophage field, and so it is okay to have some information on the broader diversity of the macrophages, but it could be condensed as the focus of the paper is not macrophage diversity. 

 Section 5:  As mentioned above, it would be good if this aligned more closely with figure 4 and the parts of the story found in figure 4 were all moved to subsections in section 5 -or even all made as their own sections. 

In terms of macrophages, M(Hb) macrophages are of course key. However, the focus here is on CD163+ macrophages, and how they associate with plaque vulnerability. While CD163 is present on M(Hb) macrophages, it is not exclusive to them; there are CD163+ cells in regions of the plaque that are not regions of haemorrhage and there are CD163+ foam cells.  I appreciate that there is no definitive marker for M(Hb) cells, and that haemorrhage is a key reason that there are CD163+ cells in the plaque, so just a bit of clarification here that not all CD163+ cells are M(hb) would be good.

Section 6: Having established the importance of IPH then this section asks the important question as to how well we can clinically identify IPH.  This is not my area of expertise, so I cannot make critical comments here.

 Minor points

·         As I am sure the authors are aware, the figure quality needs to be improved as the text is blurry.

·         Line 76, There is a spelling mistake? Glagov’s phenomenon, not Gragov’s phenomenon

·         Line 119 doesn’t read clearly: ‘Therefore, IPH is one of the critical findings as a characteristic of unstable atheroma that can lead an ultimate plaque rupture and acute coronary thrombotic occlusion’. Change highlighted wording to ‘ultimately lead to plaque rupture’

·         Note there is no reference for that statement ‘Chinetti-Gbaguidi et al.  demonstrated M2 phenotype sub-populations depending on their different activation stimuli’.  If it is meant to be the reference that the figure (figure 3) comes from (ref 50), that is inappropriate, as it is a review paper.  If you meant ‘describes’ or ‘outlines’, rather than ‘demonstrated’, then it may just be that the word demonstrated needs to be replaced. But also the figure goes beyond M2, so ’M2’ needs to go as well. 

Reviewer 2 Report

The manuscript describes in great detail the role of plaque hemorrhage for vulnerable plaques.

I miss more details for the hemodynamic effects even if that was not the topic.

In Figure 4 the schematic drawing of the vessel in a cross-sectional cut is wrong and misleading. The design of  protrusion of the plaque with NC and hemorrhage into the lumen does not exist based on pathological anatomical publications. Please have a look to the original work of Stary at al and Glagov et al.

In many intravascular ultrasound as well as OCT view the correct view to plaque formation in vivo is imaged. The circular open lumen is provide for a very long time due to remodelling. 

Round 2

Reviewer 1 Report

I thank the authors for considering my recommendations and taking them on board. I am happy with the changes made and feel that the flow of the revised manuscript works well. 

Reviewer 2 Report

The revised version is perfect. I congratulate to the revised figure, which looks now as it should be